# Benchmarking Foundation Models with Language-Model-as-an-Examiner

**Yushi Bai**[1*], **Jiahao Ying**[2*], **Yixin Cao**[2], **Xin Lv**[1], **Yuze He**[1],
**Xiaozhi Wang**[1], **Jifan Yu**[1], **Kaisheng Zeng**[1], **Yijia Xiao**[3],
**Haozhe Lyu**[4], **Jiayin Zhang**[1], **Juanzi Li**[1], **Lei Hou**[1⊠]
[1]Tsinghua University, Beijing, China    [2]Singapore Management University, Singapore
[3]University of California, Los Angeles, CA, USA
[4]Beijing University of Posts and Telecommunications, Beijing, China

## Abstract

Numerous benchmarks have been established to assess the performance of foundation models on open-ended question answering, which serves as a comprehensive test of a model's ability to understand and generate language in a manner similar to humans. Most of these works focus on proposing new datasets, however, we see two main issues within previous benchmarking pipelines, namely testing leakage and evaluation automation. In this paper, we propose a novel benchmarking framework, Language-Model-as-an-Examiner, where the LM serves as a knowledgeable examiner that formulates questions based on its knowledge and evaluates responses in a reference-free manner. Our framework allows for effortless extensibility as various LMs can be adopted as the examiner, and the questions can be constantly updated given more diverse trigger topics. For a more comprehensive and equitable evaluation, we devise three strategies: (1) We instruct the LM examiner to generate questions across a multitude of domains to probe for a broad acquisition, and raise follow-up questions to engage in a more in-depth assessment. (2) Upon evaluation, the examiner combines both scoring and ranking measurements, providing a reliable result as it aligns closely with human annotations. (3) We additionally propose a decentralized Peer-examination method to address the biases in a single examiner. Our data and benchmarking results are available at: `http://lmexam.xlore.cn`.

## 1 Introduction

Recently, many large foundation models [1], such as ChatGPT [2], LLaMA [3], and PaLM [4], have emerged with impressive general intelligence and assisted billions of users worldwide. For various users' questions, they can generate a human-like response. However, the answers are not always trustworthy, e.g., hallucination [5]. To understand the strengths and weaknesses of foundation models, various benchmarks have been established [6, 7, 8, 9, 10].

Nevertheless, we see two main hurdles in existing benchmarking methods, as summarized below. **(1) Testing leakage**. Along with increasing tasks and corpus involved in pre-training, the answer to the testing sample may have been seen and the performance is thus over-estimated. **(2) Evaluation automation**. Evaluating machine-generated texts is a long-standing challenge. Thus, researchers often convert the tasks into multi-choice problems to ease the quantitative analysis. This is clearly against real scenarios — as user-machine communications are mostly open-ended Question Answering (QA) or freeform QA [11]. On the other hand, due to the existence of a vast number of valid "good" answers, it is impossible to define one or several groundtruth, making similarity-based matching

---

*Equal contribution

37th Conference on Neural Information Processing Systems (NeurIPS 2023) Track on Datasets and Benchmarks.

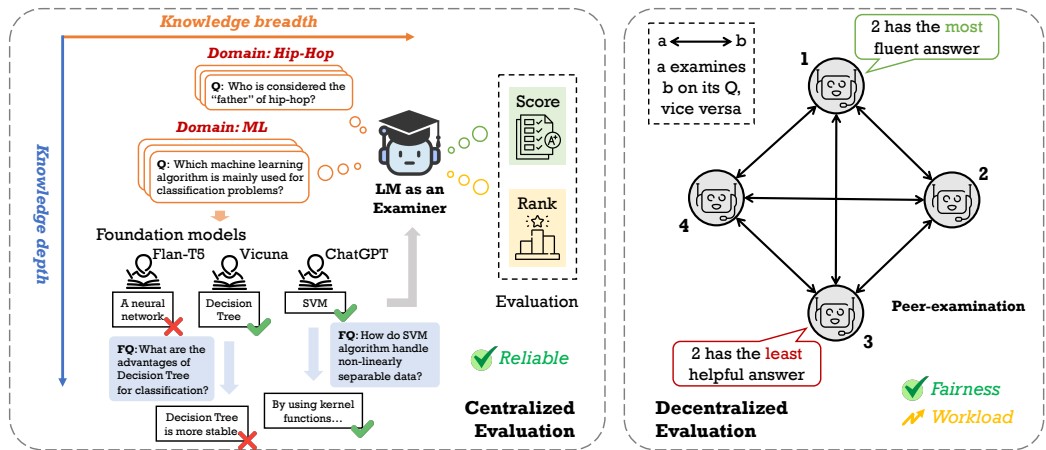

Figure 1: Overview of our benchmarking method. The left part shows the use of language model as an examiner. The examiner generates questions from various domains, allowing it to probe for comprehensive understanding (knowledge breadth) as well as deep specialization (knowledge depth) through follow-up questions (FQs). It then scores and ranks other models' responses according to its understanding of the subject, providing a reliable evaluation. The right part presents peer-examination, a novel decentralized method that provides fairer evaluation results, which potentially demands higher workload of running multiple LM examiners, compared to running a single LM examiner.

measurements (e.g., Exact Match, ROUGE-L [12], and BERTScore [13]) ineffective [11, 14, 15]. Therefore, recent works target a well-trained evaluator language model (LM) to assess the answer quality in a reference-free manner [16, 17, 18]. However, using LM as an evaluator also presents a problem: What if the evaluator hallucinates and makes wrong judgments during assessment?

As an attempt, our pilot study utilizes GPT-4 [19] to evaluate the correctness of LLaMA [3] on *Natural Questions* [20], where non-negligible 18 out of 100 judgments are incorrect (cases in Appendix A). We attribute the main reason to the inadequate knowledge of the evaluator itself regarding the questions. A straightforward solution is to use the LM not just as an evaluator to assess the responses, but as a knowledgeable examiner to also formulate questions, which is guaranteed a thorough understanding of the judgments. And, it naturally addresses the testing leakage issue by generating new questions periodically. Yet, relying on a centralized examiner can hardly be considered fair, especially when evaluating the examiner itself — *A man who is his own lawyer has a fool for his client*.

In this paper, we propose a novel benchmarking framework, *Language-Model-as-an-Examiner*, to assess current foundation models, mitigating the aforementioned issues. Herein, the language model acts as a knowledgeable examiner that poses questions based on its inherent knowledge and evaluates others on their responses. We devise three strategies to alleviate potential bias:

- **Increasing Knowledge Breadth and Depth**. In terms of breadth, according to a predefined taxonomy, we select as many diverse domains as possible to generate questions. In terms of depth, to probe models deeply within a specific subfield, we propose a multi-round setting where the evaluator mimics an interviewer, posing more sophisticated follow-up questions based on the interviewee model's preceding responses. We release our dataset, namely LMExamQA, which is constructed using GPT-4 [19] as an examiner.
- **Reliable Evaluation Measurement**. We explore two evaluation metrics, namely Likert scale scoring and Ranking, offering a more comprehensive evaluation result. The results from both metrics correlate closely with human annotations, significantly outperforming all previous metrics.
- **Peer-examination Mechanism**. To avoid the potential bias arising from a single model as examiner, we propose a decentralized evaluation setting where all participating models are invited to be the examiner and assess each other.

In experiments, our benchmarking pipeline yields fruitful results on 8 popular foundation models. We also demonstrate that peer-examination can generate a more diverse set of questions for knowledge probing and balance the biases from individual evaluator models, ultimately leading to a more equitable evaluation outcome.

## 2 Related Work

**Benchmarks for Foundation Models**. Various benchmarks have been proposed to assess foundation models on open-ended question answering, since it is the most natural setting for user-machine interaction in real scenarios. Some prominent such benchmarks include MS MARCO [21], SQuAD [22, 23], Natural Questions [20], WebQuestions [24] and OpenBookQA [25]. On the other hand, there exist a limited number of datasets that feature long-form QA. One of the widely-recognized examples is ELI5 [26], which comprises questions that necessitate lengthy descriptive and explanatory answers. One notable limitation of these benchmarks is their reliance on human curation and annotation, which inherently constrains their scalability. Our approach, by comparison, utilizes LMs to construct datasets, offering the advantage of effortless extensibility.

**Automating NLG Evaluation**. To evaluate machine-generated responses to the questions, several automatic metrics have been adopted, including the F1 score, Exact Match (EM), BLEU [27], ROUGE [12], and METEOR [28]. However, each metric has its own shortcomings, resulting in large discrepancies between the tested and actual performance [14, 29, 30].

To address these issues, well-trained LMs are utilized in NLG evaluation [31, 32, 33, 34]. One mainstream of previous methods is *reference-based*, where they derive the similarity between the candidate and the reference using an LM. Some prominent metrics in this class include Mover-Score [35], BERTScore [13]. These metrics measure the distributional similarity rather than lexical overlap [36], making them appropriate for contexts that require more flexible generation. Recent studies [16, 17, 18, 37, 38, 39, 40, 41] have demonstrated that large language models (LLMs), such as ChatGPT [2], can conduct NLG evaluations in a *reference-free* manner. They can rate a candidate text (or perform a comparative assessment of two candidates) based on a specified evaluation aspect, displaying a high correlation with human assessments in tasks such as summarization and story generation [42, 43]. In these studies, the evaluations primarily focus on lexical quality aspects, such as coherence and fluency, of a generated text. However, their capability to evaluate crucial aspects in a QA response, including factual correctness and information comprehensiveness, remains uncertain. Moreover, a single evaluator inevitably brings bias to the assessment [17]. Our work aims to resolve these issues by leveraging LM not just as an evaluator but also as an examiner, assessing the performance of other models through self-generated questions, and deploying multiple LM examiners to ensure balanced evaluation.

## 3 Methodology

In this section, we discuss the methodology in language-model-as-an-examiner, including the LMExamQA dataset construction, the evaluation metric design, and the peer-examination pipeline.

### 3.1 Dataset Construction

**Question Generation towards Knowledge Breadth**. We employ a language model (LM) as an examiner that generates diversifying and high-quality questions across various domains. To ensure wide coverage of knowledge, we choose the Google Trends Categories [2] as the domain taxonomy, and randomly select $n$ domains from it. For each domain, we prompt the LM to generate $m$ distinct questions. Our designed prompt (shown in Appendix B) is formulated to ensure that the generated questions possess three essential characteristics: diversified question forms, varied cognitive levels, and most importantly, assurance that the LM has a comprehensive understanding of the knowledge surrounding the question it poses. Figure 2 shows the distribution of question forms based on their interrogative words, and the distribution of question domains. According to Bloom's taxonomy [44], we divide the questions into 3 categories based on their required cognitive levels, from low to high-level, namely *knowledge memorization*, *knowledge comprehension*, and *knowledge analysis*:

- **Knowledge memorization**. Questions of such level demand recognition or recollection of certain entities and attributes, such as a person, location, or time.
- **Knowledge comprehension**. These questions involve demonstrating an understanding of particular instances or concepts, such as "What is . . . ", "Why . . . ", and "How . . . ".

---

[2]`https://github.com/pat310/google-trends-api/wiki/Google-Trends-Categories`.

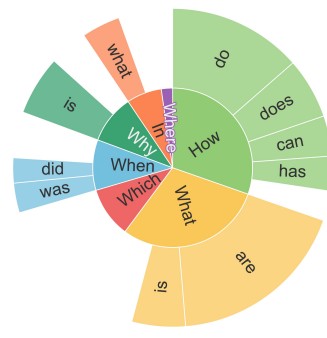

(a) Question word distribution.

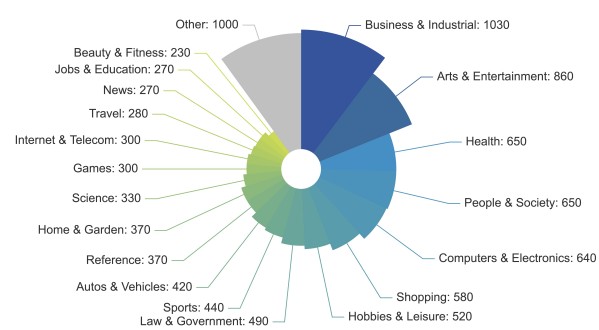

(b) Question domain distribution.

Figure 2: Statistics of generated questions in LMExamQA.

|  | | MS [21] | SQuAD2.0 [23] | NQ [20] | ELI5 [26] | Ours | Example questions in our dataset |
|---|---|---|---|---|---|---|---|
| | **Analysis** | 1% | 4% | 3% | 0% | 35% | What are the potential short and long-term impacts of divorce on children? |
| | **Comprehension** | 4% | 13% | 19% | 100% | 43% | How does towing capacity affect a truck's performance and what factors influence its maximum towing limit? |
| | **memorization** | 95% | 83% | 78% | 0% | 22% | Which international organization publishes the World Economic Outlook report? |

Table 1: Proportions of each level of questions. MS and NQ are short for MS MARCO and Natural Questions. We also list an example question in LMExamQA for each category.

- **Knowledge analysis**. Questions of this type require more advanced cognitive skills and they typically question the impact, comparison, or advantages and disadvantages of a given topic.

By adopting GPT-4 to categorize the questions in LMExamQA and previous open-ended QA datasets into three levels [3], we obtain the distribution with respect to the 3 cognitive levels as listed in Table 1, and show an example for each type of question. Compared with previous datasets, LMExamQA achieves a more balanced distribution across these 3 levels, thus providing a means of quantifying foundational models' proficiency at each cognitive level. Furthermore, LMExamQA includes a larger proportion of questions classified within higher cognitive levels, particularly at the analysis level, indicating a greater level of challenge.

To justify the reliability of the LM examiner as an evaluator on these questions, we employ it to produce a groundtruth answer with the prompt, "Answer the questions accurately and completely, without providing additional details." Upon evaluation by human experts on a random selection of 100 questions, the answers offered by the LM exhibit a $100\%$ accuracy rate, thereby demonstrating mastery over the questions it generates.

**Multi-round Follow-up Question Generation towards Knowledge Depth**. To further probe the model's comprehension of a topic in depth, we develop an evaluation procedure involving multiple rounds of follow-up inquiries, drawing inspiration from the interview process. We utilize the LM examiner to construct a series of follow-up inquiries (prompt is shown in the Appendix B). These follow-up questions are specifically tailored to delve deeper into the concepts presented within the model-generated answers from the previous round. As the follow-up questions are dependent on the model's generated answers, we only ask follow-up questions for the correctly answered queries (determined by the LM examiner) and calculate the proportion of correct responses in the subsequent round. We limit the total number of rounds to $k$ in order to minimize topic deviation that might occur during longer sessions. Note that we only provide the interviewee model with the follow-up question as input, rather than engaging the "exam history" [4], since most models are not capable of multi-round conversations. We show an example of a follow-up question to Flan-T5 [48]:

---

[3]We manually label 100 of the questions in LMExamQA and find a high agreement (85%) between human and GPT-4 annotations.

[4]It is important to note that our approach is essentially different with conversational QA [45, 46, 47], which places greater emphasis on evaluating the model's comprehension of the conversational context.

> **Question:** Which material is primarily used to manufacture semiconductor devices?
> **Flan-T5:** Silicon ✓
> **Follow-up Question:** What are the advantages of using silicon as the primary material for semiconductor devices?
> **Flan-T5:** Silicon is a nonrenewable resource, and it is the most abundant element on Earth. ✗

## 3.2 Evaluation Metrics

Several methodologies are commonly employed to facilitate human-like evaluation in LMs, prominent among these are the Likert scale scoring [16, 17, 41] and pairwise comparison [38, 41]. For the purposes of our benchmark, we incorporate both Likert scale scoring and a variant of pairwise comparison, namely ranking.

Likert scale scoring functions as an absolute evaluative measure, where the evaluator assigns scores to a given response along predefined dimensions. We establish four distinct dimensions on our dataset: (1) Accuracy. This assesses the extent to which the provided response accurately answers the question. (2) Coherence. This evaluates the logical structure and organization of the response and the degree to which it can be comprehended by non-specialists. (3) Factuality. This examines whether the response contains factual inaccuracies. (4) Comprehensiveness. This gauges whether the response encompasses multiple facets of the question, thus providing a thorough answer. Each of these dimensions is scored on a scale of 1 to 3, ranging from worst to best. We also ask the evaluator to provide an overall score ranging from 1 to 5, based on the scores assigned to the previous 4 dimensions. This score serves as an indicator of the overall quality of the answer.

On the other hand, pairwise comparison operates as a relative evaluation method and is often more discerning compared to scoring. In this process, evaluators are given two responses and are tasked with determining which is superior, taking into account their accuracy, coherence, factuality, and comprehensiveness. Given that there are $n$ contestant models, we implement a merge sort algorithm to *rank* the $n$ responses, involving $\mathcal{O}(n \log n)$ pairwise comparisons.

## 3.3 Decentralized Evaluation: Peer-Examination

We introduce a novel decentralized method that incorporates multiple models to serve as examiners, namely Peer-examination (illustrated in the right part of Figure 1), since relying only on one centralized model as the examiner introduces the following potential drawbacks to the benchmarking process. **(1) Coverage of generated questions**: The examiner may not have a holistic understanding of certain domain knowledge. As a result, the examiner may struggle to propose questions that examine in detail on these areas, which in turn renders the scope of generated questions insufficient. **(2) Potential bias during evaluation**: The model itself may have a bias during evaluation. The bias can manifest as a preference for certain types of responses or a predisposition towards perspectives irrelevant to the quality of the responses, such as response length or linguistic style. For example, [17] shows that GPT-4 [19] prefers ChatGPT [2] summaries compared to human-written summaries. Such biases may result in unfair ranking assessment outcomes.

To mitigate these issues, during peer-examination, each model is assigned the role of an examiner separately. As examiners, they are responsible for posing questions and evaluating the answers provided by the other models. We then combine the evaluation results from each of these models by voting, and obtain a final result. This approach leverages the collective expertise and diverse perspectives of all models to improve the coverage of questions as well as ensure fairer assessments.

## 4 Experiments

To demonstrate the effectiveness of our Language-model-as-an-examiner framework, we first employ GPT-4 [19] as the examiner for a centralized evaluation, since it exhibits a broad understanding of knowledge [9, 49, 50] and a precise judgmental ability [16, 17]. In peer-examination, we also employ Claude (Claude-instant) [51], ChatGPT [2], Bard [52], and Vicuna-13B [38] as LM examiners.

|  | ROUGE-1 | ROUGE-2 | ROUGE-L | BLEU | BERTScore | GPT-4 |
|---|---|---|---|---|---|---|
| *Overall Score* | | | | | | |
| Spearman ($\rho$) | 0.197 | 0.211 | 0.158 | 0.318 | 0.219 | 0.633 |
| Kendall ($\tau$) | 0.147 | 0.159 | 0.118 | 0.241 | 0.164 | 0.554 |
| *Pairwise comparison* | | | | | | |
| Accuracy | 0.473 | 0.533 | 0.530 | 0.437 | 0.487 | 0.853 |

Table 2: LM examiner's correlation with human annotations, compared with previous metrics.

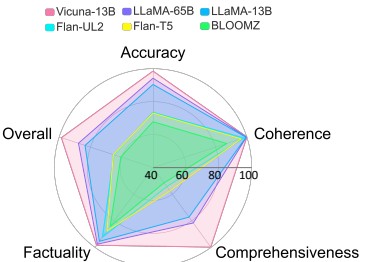

| | Single (0-shot / 5-shot) | | | | Multi |
|---|---|---|---|---|---|
| | All | memorization | Comprehension | Analysis | All |
| *Models without SFT* | | | | | |
| BLOOMZ [53] | 20.7 / 25.5 | 63.5 / 58.5 | 13.7 / 23.2 | 3.8 / 8.6 | 17.5 |
| Flan-T5 [54] | 17.0 / 26.0 | 49.8 / 62.7 | 11.2 / 21.4 | 4.6 / 9.7 | 15.1 |
| Flan-UL2 [54] | 15.5 / 24.8 | 51.3 / 64.4 | 9.2 / 20.1 | 2.2 / 7.0 | 17.0 |
| GLM-130B [55] | 14.9 / 33.3 | 45.9 / 60.7 | 9.1 / 31.9 | 3.7 / 18.7 | 18.6 |
| LLaMA-13B [3] | 29.5 / 54.3 | 72.3 / 83.9 | 23.7 / 53.2 | 11.2 / 38.1 | 43.1 |
| LLaMA-65B [3] | 38.5 / 62.9 | 78.2 / 88.2 | 34.0 / 62.7 | 20.4 / 48.1 | 53.2 |
| *Fine-tuned Models* | | | | | |
| Vicuna-13B [38] | 96.5 | 89.9 | 98.27 | 98.3 | 92.4 |
| ChatGPT [2] | 99.3 | 97.7 | 99.71 | 99.8 | 100 |

Table 3: Percentage (%) of full-mark answers on LMExamQA. We show the 0-shot and 5-shot performance for models without SFT, with both results being separated by "/".

Figure 3: Scores on each aspect. We take the 5-shot performance for models without SFT.

## 4.1 Metric Evaluation

To verify the reliability of our method for scoring and comparison based assessment, we perform metric evaluation. We conduct human evaluations on machine-generated responses. These evaluations are quantified using a 1-5 Likert scale for the overall score, and we let annotators to rank different responses for each question based on their holistic quality. We collect 300 annotations across 100 questions from LMExamQA. For each question, we randomly select 3 of the model responses, and obtain 3 scoring annotations and 3 pairwise comparison results. For Likert scoring, we calculate Spearman's $\rho$ and Kendall's $\tau$ between the overall scores given by the automatic metrics and human experts; for ranking, we compute the accuracy of pairwise comparisons offered by the automatic metrics, according to the human-labeled comparison results. Then we compare the LM examiner, GPT-4 [19], with previous automatic metrics, including ROUGE-1, ROUGE-2, ROUGE-L [12], BLEU [27], BERTScore [13] (F1), and report their correlation with human judgments in Table 2. We observe that employing GPT-4 [19] as an examiner results in a much higher correlation with human annotations compared to prior metrics. More profoundly, GPT-4's pairwise comparison achieves an agreement of over 85% with human's.

## 4.2 Centralized Benchmarking Results

**Experiment Setup**. We conduct a centralized benchmarking with GPT-4 [19] as the examiner. Following the method in Section 3, we construct the LMExamQA dataset with GPT-4, where we set $n = 1,000$ domains and $m = 10$ questions for each domain, resulting in a total of $10,000$ questions. We evaluate 8 popular and open-access foundation models on our LMExamQA dataset, including BLOOMZ (the 176B model) [53], Flan-T5 (the XXL model, 11B) [54], Flan-UL2 (20B) [54], GLM-130B [55], LLaMA (the 13B model and the 65B model) [3], Vicuna-13B [38], and ChatGPT [2]. These models are categorized based on their training procedure: whether they have undergone Supervised Fine-Tuning (SFT) or not. The first 6 models are trained without SFT [5], whereas the last 2 models are fine-tuned. For models without SFT, we assess their 0-shot and 5-shot performance. During generation, for the examiner and the subject models, we set the temperature to 0 for reproducibility. More details for reproducing our results are shown in Appendix C.2.

**Single-round QA for Knowledge Breath**. Table 3 presents the percentage of full-mark answers for each model on LMExamQA. Full-mark answers are defined as responses that receive a rating of 5 on the overall score, and the proportion of such responses is reported for each category of questions. Additionally, Figure 3 provides a radar plot depicting the average scores of models on 5 dimensions;

---

[5]Flan-T5 and Flan-UL2 are instruction fine-tuned, but they lack fine-tuning on more real scenario data.

We also conduct ranking evaluation over the 8 models (we only show the few-shot performance for models without SFT). In Figure 4, we visualize the ranking results via a win-rate heatmap (the $(i, j)$-th entry denotes the fraction of model $i$ wins when compared against model $j$) along with each model's average win-rate against all other models. We summarize our key findings.

**1. The scaling law on LMExamQA.** LLaMA-65B significantly outperforms LLaMA-13B across all question categories, adhering to the scaling law of LMs [4, 56].

**2. Few-shot leads to more substantial improvement on higher cognitive-level questions.** For models without SFT, we observe that 5-shot examples yield an average relative improvement of 17%, 123%, and 206% on memorization, comprehension, and analysis type questions, respectively. This implies that the model may possess adequate knowledge to answer higher-level questions (e.g., distinguishing between two concepts). However, it may lack the ability to retrieve knowledge from its memory and structure appropriate language to form an answer. Few-shot examples serve to provide demonstrations on how to answer such questions.

**3. What does SFT offer?** We notice a huge performance gap between LLaMA-13B and Vicuna-13B (Vicuna-13B is fine-tuned on LLaMA-13B with 70k user-shared ChatGPT conversations), mainly on the latter two types of questions. This result suggests that SFT primarily plays a crucial role in aligning LM's responses for task adaptation, rather than enriching the model's knowledge — especially in the context of higher-level questions that demand more sophisticated answers.

**4. LLMs can provide factually correct and coherent responses, but struggle for more comprehensive accurate answers.** The radar plot reveals that all models achieve relatively high scores concerning factuality and coherence (over 80/100), but different models vary widely in terms of comprehensiveness, i.e., whether the response addresses all aspects of a question.

**5. Ranking results interpretation.** Fine-tuned models, including Vicuna and ChatGPT, demonstrate near-perfect performance in terms of their scores (Table 3). In our dataset, ranking proves to be a more discerning evaluation approach. For example, the win-rate heatmap 4 reveals that ChatGPT outperforms Vicuna-13B with a 68% win rate, indicating a notable difference in the quality of responses generated by the two models. A ranking or comparison based evaluation is rarely used in QA evaluation, we encourage the research community to adopt more deliberate evaluation

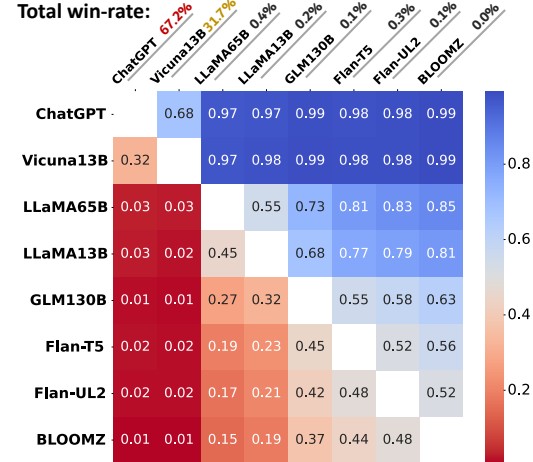

Figure 4: Win-rate heatmap under GPT-4 as an examiner.

techniques in benchmarking more advanced foundation models on open-ended QA.

**Multi-round QA for Knowledge Depth**. To conduct the multi-round QA, we randomly select 1,000 question-and-answer from the full mark answers in the first round. We then engage the examiner GPT-4 to generate the second-round question and ask the examinee models to answer the second round questions. We limit the number of rounds to $k = 2$ due to the high cost of API usage. The evaluation results are presented in the last column of Table 3. We observe that excluding ChatGPT and Vicuna-13B, all examinee models exhibit a notable decrease in performance in the second round. This suggests that while these models initially demonstrated a robust understanding and knowledge base, their performance deteriorated when faced with more complicated questions, highlighting the importance of conducting more in-depth evaluations during QA to thoroughly assess the models' capabilities. We provide more insights on the experimental results in Appendix C.2.

### 4.3 Peer-Examination Results

For the Peer-examination process, we choose four prominent models, including ChatGPT [2], Claude [51], Vicuna-13B [38], Bard [52], which are carefully selected based on their capabilities to generate questions and assess NLG quality. Each of these models is assigned the role of an

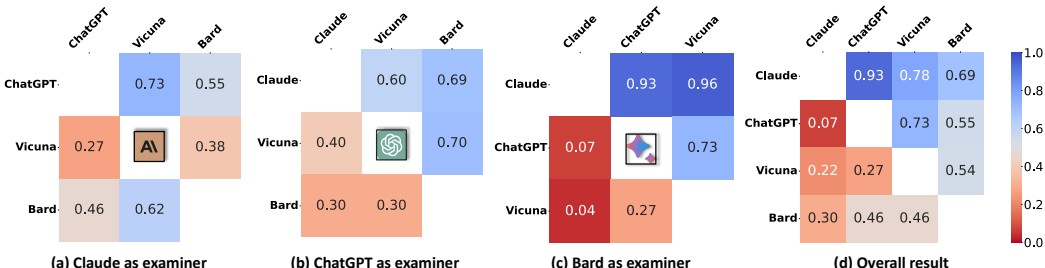

Figure 5: Win-rate heatmap under different LMs as examiners.

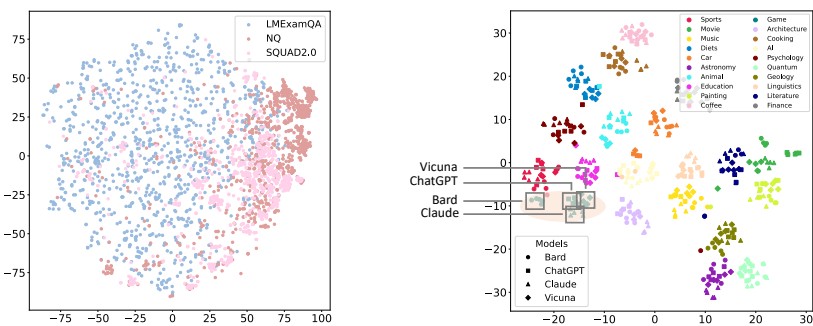

Figure 6: t-SNE on query embeddings. Left figure visualizes the embeddings of questions (generated by a centralized GPT-4 examiner) in LMExamQA; Right figure shows the embeddings of questions generated by 4 peer examiners.

examiner, posing 100 questions[6] according to the given 20 domains and evaluate the remaining three models' responses. We show the scoring results in Table 4 and the pairwise comparison results in Figure 5 (more experimental details are shown in Appendix C.3). We observe the overall rank, from highest to lowest, as follows: Claude, ChatGPT, Bard, and Vicuna-13B. Intriguingly, this aligns with the rank obtained from the popular leaderboard using the Elo rating system [57]. Our approach differs as we utilize LMs as evaluators instead of human judges.

### 4.4 Bias Analysis: Centralized vs Decentralized

We identify two potential biases in a centralized examination: one that originates from biases inherent in the questions generated by the model, and the other one rooted in the model's evaluation process.

**Bias in Generated Questions**. To analyze the bias in the generated questions, we employ t-SNE to visualize the distributions of questions across three datasets: LMExamQA, Natural Questions [20], and SQuAD2.0 [23]. These questions are encoded into 1,536-dimensional vectors using the OpenAI text-embedding model, text-embedding-ada-002 [58]. As shown in the left figure in Figure 6, we randomly select 1,000 questions on each dataset and visualize their respective t-SNE embeddings.

Through the embedding visualization, we observe that the questions in our LMExamQA dataset exhibit a more uniform distribution compared to those in previous datasets. Furthermore, we utilize 4 different LMs to generate questions across 20 domains and depict their respective question embeddings in the right panel of Figure 6. As we expected, questions within the same domain cluster together. More notably, questions produced by different models exhibit distinct distributions around the central region of the domain cluster, indicating potential biases in questions generated by a single LM. This observation motivates our adoption of peer-examination that harnesses multiple models to generate a diverse and comprehensive set of questions.

**Bias in Evaluation**. To explore potential bias in the evaluation process, we devise a bias detection experiment to test whether the centralized examiner GPT-4 exhibits a preference for a particular linguistic style. This was achieved by having GPT-4 compare a pair of responses that were identical

---

[6]We limit the total number of questions due to the unavailability of API access from Bard and Claude.

| Examiner \ Examinee | Claude | ChatGPT | Bard | Vicuna | AVG / AVG$_{weight}$ |
|---|---|---|---|---|---|
| Claude [51] | - | 98 | 100 | 96 | 98.0 / 99.7 |
| ChatGPT [2] | 41 | - | 100 | 95 | 78.6 / 98.9 |
| Bard [52] | 41 | 99 | - | 92 | 77.3 / 97.8 |
| Vicuna [38] | 42 | 98 | 99 | - | 79.6 / 99.3 |

Table 4: Percentage (%) of full-mark answer from Peer-Examination. AVG is the mean score given by the three other examiners. AVG$_{weight}$ is the mean of the scaled scores, wherein the highest score within each column is adjusted to 100 for standardization purposes.

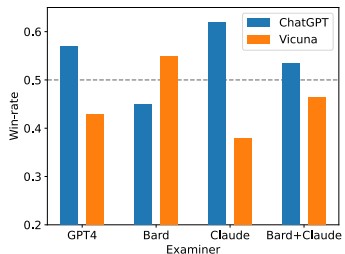

Figure 7: Win-rate of ChatGPT and Vicuna written responses under different LM examiners.

in content but varied in linguistic style. Specifically, we employ ChatGPT to rephrase the responses from Vicuna-13B. To ensure quality equivalence after paraphrasing, we ask human annotators to select 100 rewritten responses that mirror the quality of the original responses. We then present these pairs to GPT-4 for a comparative evaluation, and the results of their respective win-rate are as shown in the "GPT-4" bar in Figure 7. The results indicate that GPT-4 favor responses rephrased by ChatGPT, suggesting a potential bias towards ChatGPT style responses during the evaluation process.

To investigate whether the observed bias in the centralized examination can be mitigated using peer-examination, we incorporate two models, namely Bard [52] and Claude [51]. We instruct these models to compare the two responses, and show the results in Figure 7. The results reveal that different models possess distinct preferences. As a solution, combining them within a peer-examination framework can balance their individual biases (the rightmost bar), and lead to a more equitable evaluation process.

## 4.5  Measuring Data Leakage in Model-generated Questions

The use of model-generated questions in our framework potentially retains the risk of data leakage because the generated content by the models (e.g., the generated questions) may replicate what they have seen during pretraining. Several studies have directly and indirectly demonstrated LLMs are capable of generating creative content instead of mere replication. For instance, experiments in the GPT-2 paper [59] revealed that the median 8-gram overlap rates between GPT-2's outputs and the exact completions from the WebText test set articles were a mere 2.6%. Moreover, a line of research such as Self-Instruct [60] and Alpaca [61] demonstrates that fine-tuning models using LM-generated instructions can significantly enhance their performance. This demonstrates that under appropriate prompts, LMs are capable of generating creative content instead of replicating text encountered during training.

Nevertheless, we provide an analysis of the potential data leakage in questions generated by LM on our LMExamQA dataset. The ideal method to verify whether an LLM has encountered similar questions during training is to investigate its training data. However, the training data for most language models, often sized at several terabytes, is not openly accessible. Consequently, we investigate two primary sources of training data instead — web data and public datasets. Considering web data, we perform Google search on 100 randomly sampled questions from LMExamQA and retrieve the most similar web queries. For public datasets, we look into the Flan collection [62], which contains 1,836 tasks and 15M instances, and is widely used during instruction fine-tuning for LLMs. For each question in LMExamQA, we implement a BM25 search over the inputs from the Flan collection. Subsequently, we compute the ROUGE-L score between the question and the retrieved query. We find an average ROUGE-L score of $0.293$ and $0.046$ between the LM-generated question and the top retrieved data from the web and Flan collection, respectively. The low similarity scores in both data sources imply that the majority of the questions generated by the models are not present in the two sources. We also show 3 random questions along with their corresponding queries retrieved from web data. We can see that the model adds more qualifiers and determiners to the questions it generates than the data they've seen during pretraining, where memorizing and retelling from pretraining data is clearly not enough to answer these more precise, complex questions.

| Question | Retrieved query |
|---|---|
| How have tarot cards been adopted in modern spirituality? | What is the significance of tarot card reading? |
| What are some benefits to local economies from hosting swap meets? | How do farmers markets help the local economy? |
| How has the rise of e-commerce impacted the grocery and food retail industry? | How has e-commerce changed the retail industry? |

Table 5: Example questions and their retrieved queries.

Besides, human-collected, fixed datasets fall short in adapting to future scenarios where more pre-training data, even including data that overlaps with existing datasets, is used. With our approach, overlap in the dataset can be seamlessly addressed by re-generating the dataset via more sophisticated prompt engineering.

## 5 Conclusion

In this paper, we propose Language-Model-as-an-Examiner to address the difficulties in open-ended QA benchmarks. We construct the LMExamQA dataset, which aims to probe a more comprehensive and stratified understanding of knowledge. The dataset is equipped with a reliable language model examiner and we use it to benchmark several widely-used foundational models. To ensure a fair evaluation process, we devise a peer-examination pipeline. This approach serves to mitigate potential biases that might arise from the reliance on a single examiner. Expanding our benchmarking framework to incorporate more domain-specific language models, or even vision language models, could potentially offer a more holistic evaluation.

## 6 Limitation

Our proposed framework, despite offering a more streamlined approach for benchmarking foundation models, is not without its shortcomings. We summarize into two limitations. The first lies in the potential bias during evaluation. As we have elaborated in the paper, different models have different preferences towards distinct linguistic styles. They may also possess biases along other dimensions such as radical bias or gender bias. It's crucial to recognize these biases in future works as they might influence and be reflected in the evaluation results. Another limitation involves a lack of robust evaluation capability among existing foundation models to facilitate large-scale peer-examination. In our work, we identify only four current accessible models that demonstrate the required capacity to assess machine-generated text. We foresee in the near future that the emergence of more powerful foundation models will empower the peer-examination method, enabling more sophisticated and expansive evaluations.

**Ethical Consideration**. Creating a QA dataset using a large language model such as GPT-4 involves several ethical considerations. Here are some considerations to take into account:

- Data Privacy: Since the generated contents by LLMs may include personal information, it is crucial to ensure the anonymity of the data and protect sensitive personal information. We do not observe such information leakage in our LMExamQA dataset.
- Misinformation and Harmful Content: It is vital to ensure that the LLM generated dataset does not include or encourage misinformation, hate speech, or any form of harmful content. A rigorous review of the LMExamQA dataset assures us that such content does not appear.
- Fairness and Bias: Large language models, can unintentionally inherit and amplify societal biases present in the training data. It is important to put significant effort into identifying and mitigating such biases, as we illustrated in previous limitations.

## Acknowledgement

This work is supported by a grant from the Institute for Guo Qiang, Tsinghua University (2019GQB0003), the Tsinghua University Initiative Scientific Research Program, and the NSFC Youth Project (62006136). We gracefully thank all our anonymous reviewers for their insightful suggestions.

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

# A    Case Study in Pilot Study

In our pilot study, we utilize GPT-4 [19] to evaluate the correctness of the responses from LLaMA-13B [3] on Natural Questions [20]. For each case, we provide the question, the groundtruth (labeled in the original dataset), the response from LLaMA-13B, the exact match result, the judgment from GPT-4, and the judgment from human annotators. First, we draw attention to a case where LLaMA-13B's response is correct, yet GPT-4 incorrectly identifies it as false.

> **Question**: When were 2 dollar bills stopped being made?
> **Groundtruth**: current denomination of U.S. currency
> **Answer-LLaMA-13B**: 1966
> **Exact match**: ✗
> **GPT-4 judge**: ✗
> **Human judge**: ✓
> **Answer-GPT-4**: 2 dollar bills have not been officially discontinued, but they are printed in limited quantities and are less commonly used. ✗

Conversely, there is a case in which LLaMA-13B's answer is incorrect, but GPT-4 erroneously marks it as accurate.

> **Question**: What is the name of the first earthquake early warning system?
> **Groundtruth**: 1991 or The Mexican Seismic Alert System
> **Answer-LLaMA-13B**: ShakeAlert
> **Exact match**: ✗
> **GPT-4 judge**: ✓
> **Human judge**: ✗
> **Answer-GPT-4**: P-wave alarm system ✗

In both cases, the misjudgments by GPT-4 stem from its inadequate understanding of the knowledge related to the original questions.

# B    Example Prompts

## B.1    Question Generation

> *You have been assigned the task of developing a set of 10 different questions that demonstrate your comprehensive understanding of a specific domain. Please strictly follow these 6 rules for the task:*
>
> *1. Your questions should exhibit a thorough understanding of the domain, and should encompass both breadth and depth, incorporating different question words, such as "what", "which", "when", "where", "how", "why", etc.*
>
> *2. Make sure the first 3 questions ask about concise knowledge and can be answered in 20 words.*
>
> *3. The last 7 more complicated questions can be answered in 100 words. Among them, the last 3 questions should be compound questions.*
>
> *4. You need to generate the questions as DIVERSIFY as possible.*
>
> *5. Ensure that you can confidently answer the questions you are proposing.*
>
> *6. DO NOT add other words other than the question itself. Each question in one line, add the serial number ("1.", "2.") before each question.*
> *domain: {Domain}*

## B.2    Multi-round Question Generation

> *You have been provided with a specific domain and a question-and-answer pair related to that domain. Your task is to generate a follow-up question that delves deeper into the topic of the given question. The proposed question should be based on the answer provided in the question-and-answer pair and should aim to test the author's knowledge of the underlying concepts of the answer he proposed. To accomplish this task, please adhere to the following guidelines:*

*1. The proposed question should be closely related to the topic of the given question and should explore the same subject matter in greater detail.*

*2. You should be able to confidently answer the question you propose.*

*3. Please only return the following question as: follow question: [your proposed question].*

*Question: {Previous round question} Answer: {Previous round response}*

### B.3 Peer-Examination Question Generation

*I want you to act as a question writer expert. Your objective is to write 5 really complex and difficult questions of a specific domain to make those famous AI systems (e.g., ChatGPT and GPT-4) a bit harder to handle.*

*1. The 5 questions should be very complex and difficult, you can ask compound question.*

*2. Ensure that you can confidently answer the questions you are proposing.*

*3. DO NOT add other words other than the question itself. Each question in one line, add the serial number ("1.", "2.") before each question.*

*domain: {Domain}*

### B.4 Likert Scale Scoring

*You are a fair assessment expert, and you will be given a set of question-answer pairs. Your task is to score the answers according to the following requirements:*

*a. You should score the answer based on your knowledge of the corresponding question. You can assume your own answer to the corresponding question is the ground truth for the question.*

*b. You should rate the answer on 5 metrics, for the first 4 metrics, assign a score between 1 and 3, with 3 being the highest:*

*1. For accuracy, you will score whether the answer correctly answers the question.*

*2. For coherence, you will assess the structure and logic of the answer, and whether the answer is understandable by non-professionals.*

*3. For factuality, you will only evaluate whether the answer contains factual errors.*

*4. For comprehensive, you will determine if the answer covers multiple aspects of the question and provides a comprehensive response. For simple questions (when, which, where, etc), the plain answer itself suffices and should be rated 3.*

*5. Finally, you will provide an overall score between 1 and 5, with 5 being the highest.*

*You should only give the score, Format like: coherence: 3*

*DO NOT complete the answer!*

*Question: {Question} Answer: {Response}*

### B.5 Pairwise Evaluation

*You are a fair assessment expert, and you will be given one question along with 2 different responses. Your task is to decide which response is better. You should take into consideration the accuracy, coherence, factuality, and comprehensiveness of the responses to reach a judgment. Only return: "Response 1" or "Response 2". You do not need to explain the reason.*

*Question: {Question}*

*Response 1: {Response 1}*

*Response 2: {Response 2}*

### B.6 ChatGPT Rewrite

*You are a good writer. Paraphrase the given paragraph using more eloquent language. Include all the points and details without introducing any additional knowledge. Try to make what you write the same length as the given paragraph.*

*Paragraph: {Original paragraph}*

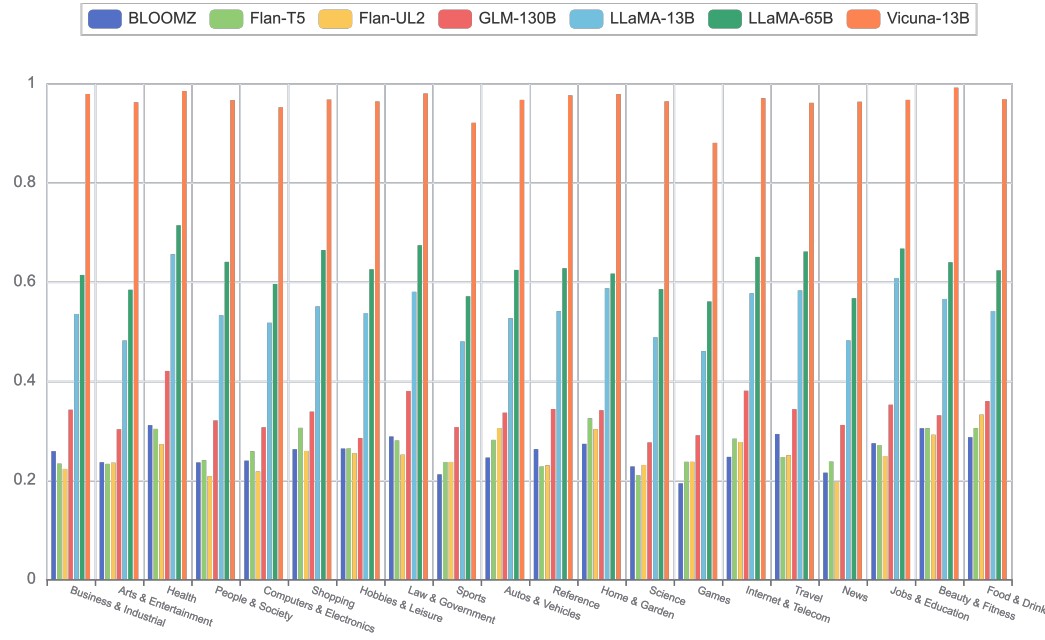

Figure 8: Percentage of full-mark answers on different domains (in the first level of the domain taxonomy) in LMExamQA. We take the 5-shot performance for models without SFT.

# C Experimental Details

## C.1 Metric Evaluation

For metric evaluation, we randomly select 100 questions from LMExamQA, and randomly select 3 model-generated responses for each question, resulting in a total of 300 samples. We ask the human annotators to score the responses between 1-5 based on their overall quality. Additionally, for each question, we ask the annotators to rank the 3 respective model-generated responses. Here we provide the annotation instructions.

1. Each column represents "question, GPT4 answer, GPT4 answer correctness, answer 1, answer 1 rating, answer 2, answer 2 rating, answer 3, answer 3 rating, answer ranking".
2. Scoring is based on five dimensions, namely accuracy (whether the question is answered correctly); coherence (whether the answer is (does it flow smoothly and is easy to understand); factuality (does it not contain factual errors); comprehensiveness (does it cover multiple aspects of the question and provide a comprehensive answer), these four dimensions are scored between 1 and 3, with 1 being the lowest and 3 the highest; the last score is a total score based on the previous evaluation dimensions, between 1 and 5, with 1 being the lowest and 5 the highest.
3. According to the above scoring rules, the last column should give the ranking of the three answers (excluding GPT4 answers) in terms of overall answer quality, in a format such as 2>3>1, representing answer 2 > answer 3 > answer 1. If there are two answers that are almost the same and hard to distinguish the better and worse, you can add "=" to the ranking, for example, 2>3=1. But please use "=" sparingly.
4. It is recommended to conduct a web search for each question to get the reference answer, and then evaluate it. It is not recommended to use the generative language models to get the reference answer. If you are really unable to evaluate on some questions (such as even using the network cannot help locate the relevant knowledge), please fill in "NULL".
5. Please refer to the first four examples for the evaluation baselines and annotation format.

## C.2 Benchmarking

**Single-round QA**. We show the evaluation results on each domain in LMExamQA in Figure 8. We provide the 0-shot and 5-shot prompts for foundation models (without SFT) to reproduce our results. We first show the 0-shot prompt for each model:

- BLOOMZ [53]:

> Question: {Question} Answer:

- Flan-UL2 [54]:

> Answer the question: {Question}

- Flan-T5 [54]:

> Question: {Question} Answer:

- GLM-130B [55] (we stop its generation once encountered a line break):

> Answer this question:
> Question: {Question}
> Answer:

- LLaMA (13B and 65B) [3] (we stop its generation once encountered a line break):

> Answer this question:
> Question: {Question}
> Answer:

To ensure fairness, all models share the same 5-shot prompts:

> Answer the following questions:
> Question: Which common household pests are controlled by professional pest control services?
> Answer: Common household pests controlled by professional pest control services include cockroaches, ants, termites, rodents, bed bugs, spiders, and wasps.
> Question: What are the key differences between assisted living and long-term care facilities?
> Answer: Assisted living facilities provide help with daily activities, social interactions, and minor medical assistance, while long-term care facilities offer extensive medical care, nursing staff support, and assistance with daily tasks for residents with serious illnesses or disabilities.
> Question: What is the primary objective of drug control policies?
> Answer: The primary objective of drug control policies is to reduce the demand, supply, and harmful consequences of illegal drugs in society.
> Question: Why is it essential to consider the type of fabric used in sleepwear when making a purchase?
> Answer: It is essential to consider the type of fabric used in sleepwear when making a purchase because it affects comfort, breathability, temperature regulation, and potential allergies or skin sensitivities.
> Question: Which historical figure is most associated with the origin of Buddhism?
> Answer: Siddhartha Gautama
> Question: {Question}
> Answer:

For the fine-tuned models, we directly enter the questions to get their answers. For all models, we set the maximum output length to 200 tokens, and the temperature to 0.

**Multi-round QA**. We also notice some notable phenomena during our multi-round QA experiment. We list them in the following.

**The knowledge pattern of foundation models is inherently different from humans.** In contrast to humans, who possess localized knowledge (e.g., knowing a person often implies knowing their actions and social connections), foundation models lack such localization. We observe that the models' accuracy remains close to that of the first round, even though they provide correct answers to the initial round questions.

**Potential deceptive results in QA capabilities for foundation models.** Our results imply that relying solely on single-round performance may lead to deceptive results, as models may simply recall previously seen texts without demonstrating genuine understanding and analysis of the knowledge.

## C.3 Peer-Examination

When conducting pairwise comparison evaluations, it is crucial to consider the issue of consistency. Specifically, the result should not change if we reverse "response 1" and "response 2".

In our test, the selected models: GPT-4 [19], Bard [52], Claude [51], and ChatGPT [2] all possess the capability to perform reliable and consistent comparison evaluations. In contrast, Vicuna-13B [38] cannot produce consistent comparison results. Hence, we employ Bard, Claude, and ChatGPT as examiners in peer-examination.

To mitigate the influence of the input sequence on the comparison evaluation process, for each comparison, we take the average of the results from the original sequence and the reversed sequence.

During the peer-review process, we need to be mindful of ChatGPT's tendency to forget (such forgetfulness is not observed for the other examiners), especially when the generated answers are too long. To tackle this, we set a maximum length threshold for responses. This limit helps make sure the comparison result from ChatGPT stays consistent. However, cutting off the answer at a certain point could remove important details or context. This could affect how accurate and high-quality the answer is. Hence we need to find a balance between limiting ChatGPT's forgetfulness and keeping the evaluation fair for every examinee model. In our experiment, to make sure the results we get from ChatGPT are reliable, we choose to slightly sacrifice fairness by setting a maximum length of 400.

**More explanations to the scoring results in Table 4.** When Claude [51] acts as the examiner, the low percentage of full-mark answers can be attributed to its strict evaluation criteria, particularly regarding the coherence dimension. This is probably because when Claude tends to provide answers containing links to related documents while exhibiting clear structures. To address the imbalance caused by Claude's strict criterion we introduce a weighted average score in addition to the arithmetic mean value. By normalizing the scores, we aim to create a more balanced comparison among the examiner models.

# D  Case Study

## D.1  Single-Round QA

### D.1.1  Knowledge Memorization Question Case

> **Field:** Beauty & Fitness → Cosmetic Procedures → Cosmetic Surgery
> **Question: What are the most common types of cosmetic surgery procedures?**
> **Ground Truth:** The most common types of cosmetic surgery procedures are breast augmentation, liposuction, rhinoplasty, eyelid surgery, and facelift.
> **Answer-ChatGPT:** The most common types of cosmetic surgery procedures include:1. Breast augmentation
> 2. Liposuction
> 3. Rhinoplasty (nose job)
> 4. Eyelid surgery
> 5. Facelift
> 6. Tummy tuck
> 7. Breast lift
> 8. Brazilian butt lift
> 9. Mommy makeover (combination of procedures to restore pre-pregnancy body)
> 10. Botox injections.
> **Score: accuracy: 3 coherence: 3 factuality: 3 comprehensive: 3 overall: 5**
> **Reason:** Accuracy: 3 (The answer correctly lists common types of cosmetic surgery procedures)
> Coherence: 3 (The answer is well-structured, easy to understand, and logically organized)
> Factuality: 3 (The answer contains no factual errors)

Comprehensiveness: 3 (The answer provides a comprehensive list covering various types of cosmetic surgery procedures) Overall Score: 5 (The answer meets all the required criteria and provides a complete and clear response to the question)

**Answer-Flan-UL2:** Breast augmentation, eyelid surgery, and rhinoplasty are the most common cosmetic surgery procedures.

**Score: accuracy: 3 coherence: 3 factuality: 3 comprehensive: 3 overall: 5**

**Reason:** Accuracy: 3 - The answer correctly lists some of the most common cosmetic surgery procedures.

Coherence: 3 - The answer is well-structured, logical, and can be understood by non-professionals.

Factuality: 3 - The answer contains no factual errors.

Comprehensiveness: 2 - The answer mentions some common procedures but could have included a few more to be more comprehensive, such as liposuction or tummy tuck.

Overall Score: 4 - The answer is accurate, coherent, and factual but could have been slightly more comprehensive.

**Answer-BLOOMZ:** The most common types of cosmetic surgery procedures include Botox injections, liposuction, and plastic surgery.

**Score: accuracy: 2 coherence: 3 factuality: 2 comprehensive: 2 overall: 3**

**Reason:** Accuracy: 2 - The answer lists some common cosmetic procedures, but Botox injections are not a surgery and plastic surgery is a general term, not a specific procedure.

Coherence: 3 - The answer is well-structured, logical, and can be understood by non-professionals.

Factuality: 2 - The answer contains a factual error by including Botox injections as a surgery and using the general term "plastic surgery."

Comprehensiveness: 2 - The answer mentions some common procedures but could have included more specific and accurate examples, such as breast augmentation, eyelid surgery, or rhinoplasty.

Overall Score: 3 - The answer is coherent but has some inaccuracies and could be more comprehensive.

**Rank: ChatGPT > Flan-UL2 > BLOOMZ**

### D.1.2 Knowledge Comprehension Question Case

**Field:** News → Health News → Health Policy

**Question: How do public health emergencies such as pandemics influence changes in health policies?**

**Ground Truth:** Public health emergencies such as pandemics influence changes in health policies by highlighting gaps in existing systems, prompting the development of new strategies, and accelerating the implementation of necessary measures to prevent, control, and mitigate the impact of the emergency.

**Answer-ChatGPT:** Public health emergencies such as pandemics can have a significant impact on health policies. Here are some ways in which pandemics can influence changes in health policies:

1. Increased funding for public health: Pandemics can highlight the importance of investing in public health infrastructure and preparedness. Governments may increase funding for public health agencies and research to better prepare for future outbreaks.

2. Changes in healthcare delivery: Pandemics can lead to changes in healthcare delivery, such as increased use of telemedicine and remote consultations. This can help to reduce the spread of infectious diseases and improve access to healthcare for vulnerable populations.

3. Development of new policies: Pandemics can lead to the development of new policies to address specific issues related to the outbreak. For example, policies may be developed to ensure that healthcare workers have access to personal protective equipment (PPE) or to mandate social distancing measures.

4. Increased focus on prevention: Pandemics can highlight the importance of prevention in healthcare. Governments may invest in public health campaigns to promote healthy behaviors and encourage vaccination.

5. International cooperation: Pandemics can lead to increased international cooperation and collaboration on health policies.

Countries may work together to share information and resources to better respond to outbreaks and prevent future pandemics.

Overall, pandemics can have a significant impact on health policies, leading to changes in healthcare delivery, increased funding for public health, and the development of new policies to address specific issues related to the outbreak

**Score: accuracy: 3 coherence: 3 factuality: 3 comprehensive: 3 overall: 5**

**Reason:** Accuracy: 3 The answer correctly addresses how pandemics influence changes in health policies.

Coherence: 3 The answer is well-structured, logical, and easily understandable by non-professionals.

Factuality: 3 The answer does not contain any factual errors.

Comprehensive: 3 The answer covers multiple aspects of the question and provides a comprehensive response.

Overall Score: 5 The answer is accurate, coherent, factual, and comprehensive, making it deserving of the highest overall score.

**Answer-Vicuna-13B:** Public health emergencies such as pandemics can have a significant impact on health policies. They can lead to changes in policies related to disease surveillance, detection, and response, as well as changes in policies related to vaccination, public health preparedness, and healthcare delivery. Governments may also take steps to increase funding for public health programs and to coordinate with international organizations to address the crisis. Additionally, during public health emergencies, there may be changes to laws and regulations related to public health, such as quarantine and isolation measures, to help prevent the spread of disease.

**Score: accuracy: 3 coherence: 3 factuality: 3 comprehensive: 3 overall: 5**

**Reason:** Accuracy: 3 The answer correctly addresses the question by mentioning various ways in which health policies are influenced by public health emergencies such as pandemics.

Coherence: 3 The answer is well-structured and logically organized, explaining different aspects of health policies that could be affected. The answer is understandable by non-professionals.

Factuality: 3 There are no factual errors in the answer. The various policy changes mentioned are all reasonable responses to public health emergencies.

Comprehensive: 3 The answer covers multiple aspects of health policy changes – surveillance, detection, response, vaccination, preparedness, healthcare delivery, funding, coordination with international organizations, and changes to laws and regulations.

Overall Score: 5 The answer is accurate, coherent, fact-based, and comprehensive, providing an excellent overview of how public health emergencies can impact health policies.

**Answer-LLaMA-65B:** Public health emergencies such as pandemics influence changes in health policies by highlighting the need for improved healthcare infrastructure, better disease surveillance, and more effective vaccines.

**Score: accuracy: 3 coherence: 3 factuality: 3 comprehensive: 2 overall: 4**

**Reason:** Accuracy: 3 The answer correctly addresses how public health emergencies influence changes in health policies.

Coherence: 3 The answer is well-structured, logical, and easily understandable by non-professionals.

Factuality: 3 The answer contains no factual errors.

Comprehensive: 2 The answer covers some aspects of the question but could provide more details on how health policies change in response to public health emergencies.

Overall Score: 4 The answer is accurate, coherent, and factual, but could be more comprehensive in addressing the question.

**Answer-LLaMA-13B:** Public health emergencies such as pandemics influence changes in health policies by increasing the demand for healthcare services and resources.

**Score: accuracy: 2 coherence: 3 factuality: 3 comprehensive: 2 overall: 3**

**Reason:** Accuracy: 2 The answer is partially correct, as it mentions the increased demand for healthcare services and resources, but it doesn't cover other ways in which health policies may change during a pandemic, such as implementing new prevention measures or modifying existing ones.

Coherence: 3 The answer is well-structured and logical. It is easily understandable by non-professionals.

Factuality: 3 The answer is factually correct in stating that public health emergencies increase the demand for healthcare services and resources.

Comprehensiveness: 2 The answer addresses one aspect of the question but does not provide a comprehensive response by including other potential changes in health policies during a pandemic.

Overall Score: 3 The answer is partially correct and understandable, but it could be more comprehensive in addressing other changes in health policies.

**Rank: ChatGPT > Vicuna-13B > LLaMA-65B > LLaMA-13B**

### D.1.3 Knowledge Analysis Question Case

**Field:** Internet & Telecom → Service Providers → Cable & Satellite Providers

**Question: What are the advantages and disadvantages of bundling services like internet, television, and phone from a single provider in the context of pricing and service quality?**

**Ground Truth:** Advantages: 1. Cost savings: Bundling services often leads to discounts and lower overall costs.

2. Convenience: Dealing with a single provider simplifies billing and customer service.

Disadvantages: 1. Limited options: Bundling may restrict choices for individual services, leading to less customization.

2. Service quality: A single provider may not excel in all services, potentially compromising quality in some areas.

**Answer-ChatGPT:** Advantages of bundling services:

1. Cost savings: Bundling services can often result in cost savings for customers as providers offer discounts for multiple services.

2. Convenience: Having all services from a single provider can be convenient as customers only need to deal with one company for all their needs.

3. Simplified billing: Bundling services can simplify billing as customers receive a single bill for all services.

4. Additional features: Providers may offer additional features or perks for customers who bundle services, such as free premium channels or faster internet speeds.

Disadvantages of bundling services:

1. Limited options: Bundling services may limit customers' options as they may be forced to choose from a limited selection of providers.

2. Service quality: Bundling services may result in lower service quality as providers may prioritize certain services over others.

3. Contract terms: Bundling services often requires customers to sign a contract, which may have strict terms and conditions.

4. Difficulty switching providers: Bundling services may make it difficult for customers to switch providers as they would need to cancel all services and find new providers for each service.

**Score: accuracy: 3 coherence: 3 factuality: 3 comprehensive: 3 overall: 5**

**Reason:** Accuracy: 3 The answer correctly identifies the advantages and disadvantages of bundling services in the context of pricing and service quality.

Coherence: 3 The answer is well-structured, logically organized, and easily understandable by non-professionals.

Factuality: 3 The answer does not contain any factual errors and provides accurate information about the advantages and disadvantages of bundling services.

Comprehensiveness: 3 The answer covers multiple aspects of the question and provides a comprehensive response, discussing both advantages and disadvantages of bundling services.

Overall Score: 5 The answer is accurate, coherent, factual, and comprehensive, providing a thorough and well-rounded response to the question.

**Answer-GLM-130B:** Bundling services like internet, television, and phone from a single provider in the context of pricing and service quality has advantages and disadvantages. Bundling services like internet, television, and phone from a single provider in the context of pricing and service quality has advantages because it allows consumers to save money on services, and it has disadvantages because it can lead to higher prices and lower service quality.

**Score: accuracy: 3 coherence: 3 factuality: 3 comprehensive: 2 overall: 4**

**Reason:** Accuracy: 3 The answer correctly identifies the advantages and disadvantages of bundling services in the context of pricing and service quality.
Coherence: 3 The answer is well-structured, logical, and easily understandable by non-professionals.
Factuality: 3 The answer does not contain any factual errors.
Comprehensiveness: 2 The answer provides a general overview of the advantages and disadvantages but could have elaborated more on the reasons behind the cost savings and potential issues with service quality.
Overall Score: 4 The answer is accurate, coherent, and factual, but could have been more comprehensive in explaining the reasons behind the advantages and disadvantages.
**Answer-Flan-T5:** Bundling services like internet, television, and phone from a single provider can result in lower prices and better service quality.
**Score: ccuracy: 2 coherence: 3 factuality: 3 comprehensive: 2 overall: 3**
**Reason:** Accuracy: 2 The answer mentions lower prices and better service quality as advantages, but it does not mention any disadvantages.
Coherence: 3 The answer is well-structured, logical, and easily understandable by non-professionals.
Factuality: 3 The answer contains no factual errors.
Comprehensive: 2 The answer covers the advantages of bundling services but does not address the disadvantages.
Overall Score: 3 The answer provides a partial response to the question, mentioning the advantages but not the disadvantages of bundling services.
**Answer-BLOOMZ:** The advantages of bundling services like internet, television, and phone from a single provider include lower prices and better service quality, while the disadvantages include higher prices and less choice.
**Score: accuracy: 2 coherence: 3 factuality: 3 comprehensive: 2 overall: 3**
**Reason:** Accuracy: 2 The answer mentions both advantages and disadvantages, but it contradicts itself by stating both lower and higher prices as advantages and disadvantages.
Coherence: 3 The answer is well-structured and easy to understand for non-professionals.
Factuality: 2 The answer contains a factual error by stating both lower and higher prices as advantages and disadvantages. Bundling services usually result in lower prices, not higher.
Comprehensiveness: 2 The answer covers some aspects of the question but could provide more details on the advantages and disadvantages of bundling services.
Overall Score: 3 The answer is somewhat accurate and coherent but could be improved in terms of factuality and comprehensiveness.
**Rank: ChatGPT > GLM-130B > BLOOMZ > Flan-T5**

## D.2 Multi-turn Question Evaluation

**Field:** Hobbies & Leisure → Cycling → Bicycles & Accessories
**Question: Which material is most commonly used for road bike frames?**
**Answer-Flan-T5:** Aluminium is most commonly used for road bike frames. ✓
**Following-Question: What are the advantages and disadvantages of using aluminium for road bike frames compared to other materials like carbon fiber and steel?**
**Answer-Flan-T5:** Aluminium is a light, strong, and durable material that is used for road bike frames. Aluminium is also a cheap material, making it a popular choice for budget road bikes. Aluminium is also a good choice for road bikes because it is easy to weld and can be made into a variety of shapes.
**Score: accuracy: 2 coherence: 3 factuality: 3 comprehensive: 1 overall: 2**
**Reason:** Accuracy: 2 The answer mentions some advantages of using aluminium but does not discuss the disadvantages or compare it to carbon fiber and steel.
Coherence: 3
The answer is well-structured, clear, and easily understandable.
Factuality: 3
The facts presented about aluminium are correct.
Comprehensive: 1
The answer does not cover multiple aspects of the question; it only focuses on the advantages of aluminium and does not discuss disadvantages or comparisons with carbon fiber and steel.

Overall score: 2
The answer provides some accurate information about aluminium but fails to fully address the question regarding disadvantages and comparisons with other materials.

**Field:** Sports
**Question: In basketball, what defensive strategies are commonly employed to disrupt an opposing team's offensive flow?**
**Answer-GLM-130B:** In basketball, defensive strategies commonly employed to disrupt an opposing team's offensive flow include trapping, double-teaming, and switching. ✓
**Following Question: Can you describe the differences between trapping, double-teaming, and switching in basketball defense?**
**Answer-GLM-130B:** Trapping is a defensive strategy in which a defender attempts to block the path of a ball-handler by moving in front of him or her. Double-teaming is a defensive strategy in which two defenders attempt to block the path of a ball-handler by moving in front of him or her. Switching is a defensive strategy in which a defender moves from one side of the court to the other to block the path of a ball-handler.
**Score: accuracy: 2 coherence: 3 factuality: 2 comprehensive: 2 overall: 3**
**Reason:** Accuracy: 2 The answer provides a basic understanding of the three defensive strategies but lacks some important details. For example, trapping usually involves two defenders closing in on the ball-handler, not just one.
Coherence: 3 The answer is well-structured and easy to understand for non-professionals.
Factuality: 2 The answer contains some factual errors, such as the description of trapping involving only one defender.
Comprehensive: 2 The answer provides a basic understanding of the three defensive strategies but lacks some important details, such as the purpose of each strategy and how they are executed in a game.
Overall Score: 3 The answer provides a basic understanding of the three defensive strategies but could be improved with more accurate and detailed information.

**Field:** Health → Health News → Health Policy
**Question: Which organization is primarily responsible for global health policies and guidelines?**
**Answer-LLaMA-65B:** The World Health Organization (WHO) is primarily responsible for global health policies and guidelines. ✓
**Following Question Can you describe the process by which the World Health Organization develops and implements these global health policies and guidelines?**
**Answer-LLaMA-65B:** The World Health Organization develops and implements global health policies and guidelines through a process of consultation with experts, member states, and other stakeholders.
**Score: accuracy: 3 coherence: 3 factuality: 3 comprehensive: 2 overall: 4**
**Reason:** Accuracy: 3 Reason: The answer correctly describes the process of consultation with experts, member states, and stakeholders.
Coherence: 3 Reason: The answer is well-structured, logical, and easily understandable by non-professionals.
Factuality: 3 Reason: The answer contains no factual errors.
Comprehensiveness: 2 Reason: The answer provides a general overview of the process but could include more details about the specific steps involved in developing and implementing policies and guidelines.
Overall Score: 4 Reason: The answer is accurate, coherent, and factual, but could be more comprehensive in describing the process.

## D.3  Peer-Examination Evaluation

**Examiner: Bard**
**Field:** Psychology
**Question: How does attachment style influence romantic relationships?**
**Answer-Vicuna-13B:** Attachment style, which refers to the way individuals form emotional bonds with others, can have a significant impact on romantic relationships. People with a

secure attachment style tend to have healthy, stable relationships, while those with insecure attachment styles may struggle with issues such as jealousy, possessiveness, and trust problems. For example, individuals with an anxious attachment style may become overly dependent on their partner and fear abandonment, while those with a avoidant attachment style may be less committed and emotionally distant. Understanding and addressing attachment style issues can help individuals improve the quality and stability of their romantic relationships.

**Answer-ChatGPT:** Attachment style can have a significant impact on romantic relationships. People with a secure attachment style tend to have healthy and satisfying relationships, while those with insecure attachment styles may struggle with intimacy and trust.

Those with an anxious attachment style may be overly dependent on their partner and fear abandonment, leading to clingy or jealous behavior. They may also have a tendency to overanalyze their partner's actions and words, leading to misunderstandings and conflict.

Those with an avoidant attachment style may have difficulty with emotional intimacy and may prioritize independence over closeness. They may also struggle with expressing their emotions and may come across as distant or uninterested in their partner's feelings.

Overall, understanding one's attachment style and how it affects their behavior in relationships can be helpful in improving communication, building trust, and creating a more fulfilling partnership.

**Answer-Claude:** Attachment style, formed in early childhood, can significantly impact romantic relationships. Here are some of the main ways:

• Secure attachment style - People with secure attachment tend to have the healthiest relationships. They are comfortable with intimacy and vulnerability, and trust their partner. They communicate well and resolve conflicts in a constructive manner.

• Anxious-preoccupied attachment - People with this style often become overly dependent on their partner for validation and reassurance. They tend to worry excessively about the relationship and their partner's feelings. This can be smothering and drive partners away.

• Avoidant attachment - People with an avoidant style have difficulty opening up emotionally and being vulnerable. They tend to keep their partner at a distance and may withdraw during conflict. Their partners often feel unwanted, misunderstood, and frustrated.

• Fearful-avoidant attachment - This involves an unstable combination of anxiousness and avoidance. People with this style desire intimacy but also withdraw from it. They have a hard time fully trusting their partner. This often leads to relationship instability, distance, and misunderstandings.

• Mixed attachment styles - When partners have different attachment styles, it can also cause issues. The most compatible pairings tend to be between a secure and anxious style. Avoidant-anxious pairings often struggle the most.

In summary, having a secure attachment style helps one form the emotional foundation for a healthy, trusting relationship. The other styles often correlate with relationship difficulties, though with awareness, effective communication, and effort from both partners, many couples can work through attachment issues.

**Bard's Rank: Claude > ChatGPT > Vicuna-13B**

## D.4 Rephrase Case

**Question: Explain how machine learning algorithms, deep neural networks, and core AI techniques will likely evolve and make progress over the next 10-20 years, including what new approaches might fundamentally change the landscape of artificial intelligence research and development.**

**Answer-Vicuna-13B:** Machine learning algorithms, deep neural networks, and core AI techniques are likely to continue to evolve and make progress over the next 10-20 years through advancements in hardware, data availability, and algorithm development. New approaches such as reinforcement learning, transfer learning, and generative models are likely to fundamentally change the landscape of AI research and development by enabling more complex and nuanced decision-making, faster learning, and improved performance in a variety of applications. Additionally, the integration of AI with other emerging technologies such as 5G networks, edge computing, and quantum computing will likely lead to new breakthroughs in AI.

**Answer-ChatGPT Paraphrase:** The field of artificial intelligence is poised for significant advancements in the next decade or two, driven by improvements in hardware, data accessibility,

and algorithmic innovation. Machine learning algorithms, deep neural networks, and other core AI techniques are expected to continue to evolve and progress, while new approaches such as reinforcement learning, transfer learning, and generative models are likely to revolutionize the field by enabling more sophisticated decision-making, faster learning, and enhanced performance across a range of applications. Furthermore, the integration of AI with other emerging technologies, including 5G networks, edge computing, and quantum computing, is expected to unlock new opportunities and breakthroughs in AI research and development.

