# OpenReview forum: "Benchmarking Foundation Models with Language-Model-as-an-Examiner"
_NeurIPS.cc/2023/Track/Datasets_and_Benchmarks — NeurIPS 2023 Datasets and Benchmarks Poster_

### Official Review · Reviewer_zXrq · 2023-06-25

**Rating:** 7
**Confidence:** 3
**Clarity:** The paper is generally well written, …

**Strengths:**

Using an LM as an examiner is an interesting approach. The extensibility of the framework enables the adoption of various LMs as examiners, and the questions can be constantly updated with diverse trigger topics. The combination of scoring and ranking measurements improves the evaluation process and aligns closely with human annotations. The introduction of a decentralized peer-examination method helps address biases in evaluation. The paper provides a dataset and benchmarking results, enhancing reproducibility and further research in the field.

**Additional Feedback:**

- In Figure 1 left: it looks like Vicuna has a green check but its output has a red cross, which seems confusing. Is the output with a red cross supposed to be the output of Flan-T5?
- In Figure 1 right, what does “workload” refer to?

**Correctness:**

The proposed benchmarking framework and evaluation methods are appropriate for the task of assessing foundation models in open-ended question answering.
But some of the claims made in the submission appear to lack some quantitative evaluation (see the “Opportunities For Improvement” section above).

**Documentation:**

The paper mentions the availability of the dataset and benchmarking results at lmexam.com.

**Limitations:**

The authors have partially addressed the limitations of their work by proposing a decentralized peer-examination method, etc., but further discussion could be valuable (see the “Opportunities For Improvement” section above). The paper could also discuss the scalability of the proposed benchmarking framework in real-world scenarios.

**Opportunities For Improvement:**

The claims and limitations of the proposed approach could be evaluated more thoroughly. For instance, the paper acknowledges the issue of an evaluator LM hallucinating and making wrong judgments and proposes using LMs to formulate questions instead. But it seems that how much this approach actually reduces hallucination is not evaluated. It would be great to provide a quantitatively analysis regarding this.

Additionally, the authors claim that using LMs to formulate questions can reduce testing leakage. It would be great if the paper also explained how do the authors ensure the LM-generated questions are new and not influenced by testing leakage. Some concerns/questions I have are: would LMs tend to generate questions that look like what they have seen during training? Quantitatively, how much does the proposed approach reduce testing leakage?

**Relation To Prior Work:**

The paper discusses how this work differs from previous contributions. It highlights the limitations of existing benchmarking methods and proposes a new framework that overcomes these limitations. The combination of using LMs as examiners, introducing scoring and ranking measurements, and employing a decentralized peer-examination method distinguishes this work from prior research.

**Summary And Contributions:**

The authors propose a new benchmarking framework called Language-Model-as-an-Examiner to evaluate the performance of foundation models in open-ended question answering. The framework leverages a language model (LM) as an examiner to generate questions and evaluate responses in a reference-free manner. The contributions of the paper include the extensibility of the framework, the use of multiple domains and follow-up questions for comprehensive evaluation, the combination of scoring and ranking measurements for reliable results, and the introduction of a decentralized peer-examination method to address biases. The authors release a dataset and benchmarking results.

---

> ### Author Response · Authors · 2023-08-09
> **Thanks for your helpful review**
>
> Thanks for your review and detailed suggestions. Here are our responses for your questions.
>
> - The paper acknowledges the issue of an evaluator LM hallucinating and making wrong judgments and proposes using LMs to formulate questions instead. But it seems that how much this approach actually reduces hallucination is not evaluated. It would be great to provide a quantitative analysis regarding this.
>
> Our work shows that on self-generated questions, LM exhibits fewer hallucinations when acting as the evaluator, including providing more correct answers and aligning more closely with human judgment.
> First, as our pilot study suggests (line 37), GPT-4 makes 18 wrong judgments out of 100 LLaMA responses on NQ, due to erroneous perception of these questions (see Appendix A for details). In comparison, when GPT-4 addresses self-generated queries on LMExamQA, it showcases a 100\% success rate in providing accurate responses (as affirmed by our manual verification of a random selection of 100 responses, line 125).
> Moreover, on LMExamQA, GPT-4 displays a judgment consistency highly correlated to human judgment (Table 2), indicating that using LM's self-generated questions can reduce its hallucination during evaluation.
>
> - It would be great if the paper also explained how do the authors ensure the LM-generated questions are new and not influenced by testing leakage. Some concerns/questions I have are: would LMs tend to generate questions that look like what they have seen during training? Quantitatively, how much does the proposed approach reduce testing leakage?
>
> Please refer to the general response.
>
> - The scalability of the proposed benchmarking framework in real-world scenarios.
>
> Our automated dataset construction by the examiner LM enables effortless extensibility and constant question updating. This allows the benchmark to scale to cover an extensive range of topics and domains. However, generating a massive benchmark with thousands of domains and questions would incur significant computational costs for the LM examiners, and scoring and ranking evaluations by LMs can be quite time-consuming compared to standard automatic metrics.
>
> - Questions related to Figure 1
>
> In Figure 1 left, the red cross for Vicuna implies that Vicuna inaccurately responds to the follow-up question (FQ) despite correctly answering the initial query, as indicated by the green check mark.
> In Figure 1 right, the term 'workload' refers to the workload of running multiple LM examiners during peer-examinations, including more API expenses for API models, and increased computation cost for open-source models, compared to running a single LM examiner.

---

> > ### Comment · Reviewer_zXrq · 2023-08-25
> > **Thank you for your detailed response**
> >
> > Thank you for your detailed response, which addresses my questions. I have increased the rating.

---

### Official Review · Reviewer_6wH3 · 2023-07-18
**The paper presents a framework for using language models as examiners to generate questions and then as an evaluator to evaluate the answers. While there are some concerns with the automation, the overall framework is well thought of with the peer validation system to average out any errors.**

**Rating:** 8
**Confidence:** 4
**Correctness:** 1. The paper describes the problem of…

**Strengths:**

1. The framework for examiner can be extended to any language model.
2. A key finding on few-shot setting giving larger improvement for analysis questions over the remaining 2 categories is a valuable result to better evaluate higher-level understanding. These types of questions are especially hard to evaluate and this finding can help evaluation of future work in this open QA.
3. As automatic LM based evaluation can lead to biases, the work also studied biases both in examiner and the evaluator. The biases showed in evaluator with respect to linguistic preferences are countered by the proposed peer examination which is a useful technique to average out any peculiar behaviors.



**Additional Feedback:**

Minor corrections:
1. Page 3: “On the other hand, there exist a” -> “On the other hand, there exists a”.
2. Minor stylistic presentation choice: The 3 types of questions mentioned in Lines 110-115 are in a different order compared to Table 1. Presenting them in the same order is slightly more consistent for readability.


**Clarity:**

 Line 125 states that when the LM is used to both generate the questions and the answers, i.e., both as an examiner and the evaluator, the accuracy is 100%. It is unclear how the validation is conducted. If it was human evaluation, providing more details like how many data samples were examined makes this clear.


**Documentation:**

Line 134: The choice for the values of “k” for minimizing topic deviation in follow-up questions for depth is not thoroughly described (Line 253 later on describes API cost limitation). Also, if there were other values for k, how was the final value for the dataset decided?


**Ethics:**

The biases of the large models would be reflected in the questions generated by the examiner and the evaluator. Other than that, there are no specific ethical concerns brought about by this framework.

**Limitations:**

1. The strength of extensibility to different language models also has its own limitation. This framework relies on domain-specific language models in order to generate informative and sensible questions in domains such biomedical, law, finance etc.,
2. The in-depth evaluation seems to show a drop in performance for the second round. The paper describes how this kind of in-depth evaluation is therefore necessary (Line 255). However, it is unlear which aspects of the second question make them more difficult to answer.


**Opportunities For Improvement:**

1. The work shows that using LM as examiner and evaluator makes comprehensiveness take a hit compared to other aspects. While this is a useful finding, it is not clear what categories of comprehensiveness is being affected. An analysis on these categories can help define the shortcomings of the framework to lay out future improvements.
2. Certain families of LMs are better for types of questions like the ones that showed improvement with the supervised finetuning of the models such as Vicuna. With the peer examination, there is an opportunity to select different language models in the process of averaging like macro-Socratic models.


**Relation To Prior Work:**

Prior contributions of using LMs for evaluation is discussed well.

**Summary And Contributions:**

This paper addresses the issues of evaluation automation and leakage from the prior pipelines for benchmarking open-ended question answering by using language model as the examiner. The knowledge from pretrained models is used to make questions and evaluate the answers in a reference-free manner. The questions are framed in an increasing breadth on different domains based on pre-defined taxonomy using Google Trends Categories and increasing depth by asking follow up questions. The authors also observe that there is 100% accuracy for the answers generated by the same language model from which the questions are generated. For evaluation, a Ranking approach and Likert scale under the dimensions of accuracy, coherence, facutality and comprehensiveness are examined thoroughly with a high correlation. They also analyze the biases in the questions and answers. While questions are more uniformly distributed compared to earlier works, answers demonstrate biases to certain linguistic styles.

Main contributions:
1. The paper proposes using a language model as an examiner to both ask questions and evaluate the answers in a reference-free manner. The collected dataset with GPT-4 is publicly released. The proposed framework is extensible to any language model.
2. The evaluation paradigm of both the scoring and ranking brings the scores closer to human annotations.
3. The paper also proposes a new decentralized peer examination to better understand the biases in any single examiner.

---

> ### Author Response · Authors · 2023-08-09
> **Thanks for your comprehensive review**
>
> Thanks for your comprehensive review. Here are our responses on the questions.
> - The strength of extensibility to different language models also has its own limitation. This framework relies on domain-specific language models in order to generate informative and sensible questions in domains such biomedical, law, finance etc.
>
> We totally agree that the evaluation results with LM-as-an-examiner rely on the LM examiner's proficiency. Expanding our benchmarking framework to incorporate more domain-specific language models, or even vision language models, could potentially offer a more holistic evaluation.
>
> - During the in-depth evaluation, it is unclear which aspects of the second question make them more difficult to answer.
>
> In Appendix D.2, we provide some case studies on the multi-turn questions. Our observation indicates that the follow-up questions always require a more sophisticated analysis of the distinctions, implications, pros\&cons, and challenges associated with the subject matter. Due to their demand for more complex reasoning and logical organization capabilities, answering these follow-up questions poses a greater challenge.
>
> - Line 125 states that when the LM is used to both generate the questions and the answers, i.e., both as an examiner and the evaluator, the accuracy is 100\%. It is unclear how the validation is conducted.
>
> We manually annotate the accuracy of responses provided by GPT-4 to 100 GPT-4 generated questions in LMExamQA. We confirm that all the answers are indeed accurate. We will add some clarification in revision.
>
> - Line 134: The choice for the values of “k” for minimizing topic deviation in follow-up questions for depth is not thoroughly described (Line 253 later on describes API cost limitation). Also, if there were other values for k, how was the final value for the dataset decided?
>
> During our experiment, we find that $k\geq3$ rounds of follow-up questions could potentially result in topic deviation. This occurs when the LM examiner loses focus on the original topic, fixating instead on matters discussed in the last round. The value of $k$ should be set to guarantee that the topics of the questions are still consistent after a long enough chat history, and the follow-up questions gain depth and complexity compared to previous rounds.

---

> > ### Comment · Reviewer_6wH3 · 2023-08-30
> >
> > Thank you for providing additional context to the questions. The domain-specificity is a standalone to start most ideas.
> > It would be good to see categories of questions in addition to a few examples to understand the effectiveness of the second question.
> > Appreciate the detailed discussion about the value of k. I think this would be a valuable paper for the conference. I have no more questions.

---

### Official Review · Reviewer_JGap · 2023-07-23
**More justification for the rationale**

**Rating:** 5
**Confidence:** 4
**Correctness:** Refer to the Opportunities For Improv…

**Strengths:**

● The use of LLMs for question generation, instead of the traditional practice of using questions from existing datasets. Therefore, it can potentially generate a diverse set of questions with desired topics.

● The introduction of a peer-examination system where all models serve as examiners and assess each other.

**Additional Feedback:**

Refer to the Opportunities For Improvement.

**Clarity:**

● How do you obtain the proportions in Table 1? Are they from the manual categorization of questions in all these datasets?

● Line 122~125: this kind of manual check is based on how many questions? Also, does this check involve the answerability of those generated questions? More details should be given of the generated questions in this dataset.

**Documentation:**

Mostly discussed.

**Limitations:**

Not much about limitations was discussed.

**Opportunities For Improvement:**

● The authors' claimed motivation for addressing "testing leakage" seems unresolved. The use of model-generated questions potentially retains the risk of data leakage because such data leakage may happen because of the overlapping of pertaining data of different models.

● The practical utility of the peer-examination system is questionable, as there may not always be multiple models available for simultaneous evaluation.

● The models Vicuna-13B and ChatGPT exhibited almost perfect performance on the newly presented dataset, LMExamQA (based on GPT-4), suggesting that this dataset may not present a sufficient challenge to effectively assess future models, OR indicating the happening of data leakage, as mentioned in the above first point.

**Relation To Prior Work:**

Mostly discussed.

**Summary And Contributions:**

This paper presents a new framework for evaluating large language models (LLMs), namely Language-Model-as-an-Examiner. The main idea is the use of a more advanced LLM as the examiner, posing questions to other models and assessing their responses in a reference-free manner. The authors also introduce a decentralized evaluation setting where all models serve as examiners and mutually evaluate each other.

---

> ### Author Response · Authors · 2023-08-09
> **Thanks for your suggestions**
>
> Thanks for your insightful suggestions. Here is our response for each question.
> - The use of model-generated questions potentially retains the risk of data leakage
>
> Please see the general response.
>
> - The practical utility of the peer-examination system is questionable, as there may not always be multiple models available for simultaneous evaluation.
>
> As of August 8th, 2023, there are 528 open-sourced LLMs listed on [Hugging Face Open LLM Leaderboard](https://huggingface.co/spaces/HuggingFaceH4/open_llm_leaderboard). The number of LLMs continues to rise [1], with each exhibiting increasingly powerful capacities.
> This leads us to believe that the peer-examination method will be more applicable in the near future.
> Having multiple LLMs as examiners, trained and aligned using different data and methods, can yield more comprehensive evaluations, ultimately reducing the cost of manual data collection and the development of automated evaluation methods.
>
> - Some models exhibited almost perfect performance on LMExamQA (based on GPT-4), suggesting that this dataset may not present a sufficient challenge to effectively assess future models, OR indicating the happening of data leakage
>
> As we curate LMExamQA over a general domain (using trigger domain from google trend categories), the questions in this dataset are more general and closer to daily-life scenarios. Although some current large-scale language models may correctly answer these questions, as they are usually trained on a general domain, the quality of their full-mark responses can vary significantly. To address this, a ranking-based evaluation proposed in our paper serves as a more discerning and challenging evaluation metric, and can distinguish well between models with strong and weak capabilities (Figure 4).
> Moreover, our work emphasizes more on proposing LM-as-an-examiner as an automated method of building datasets, rather than proposing a single dataset.
> Our approach empowers future work to construct test datasets to benchmark models on more advanced tasks with ease.
> The analysis provided in our general response also suggests the issue of data leakage is negligible in our evaluation method.
>
> - How do you obtain the proportions in Table 1?
>
> We obtain the statistic in Table 1 by adopting GPT-4 to categorize the questions in LMExamQA and previous datasets into three levels. We also manually labeled 100 of the questions in LMExamQA and find a high agreement (85\%) between human and GPT-4 annotations. We will make sure to clarify this in revision.
>
> - Line 122-125: this kind of manual check is based on how many questions? Also, does this check involve the answerability of those generated questions?
>
> We check 100 questions for the correctness of the provided answers by the LM examiner. An additional check is performed to confirm that the questions are indeed answerable and do not require any external reference.
>
> - Not much about limitations was discussed.
>
> We discuss our limitations in Appendix E.
>
> [1]. Naveed H, Khan A U, Qiu S, et al. A Comprehensive Overview of Large Language Models[J]. arXiv preprint arXiv:2307.06435, 2023.

---

### Author Response · Authors · 2023-08-09
**A quantitative analysis of the potential data leakage for model-generated questions**

In the context of LLMs, data leakage suggests that the generated content (e.g., the generated questions in our paper) replicates what they have seen during pretraining. Several studies have directly and indirectly demonstrated LLMs are capable of generating creative content instead of mere replication. For instance, experiments in the GPT-2 paper [1] revealed that the median 8-gram overlap rates between GPT-2's outputs and the exact completions from the WebText test set articles were a mere 2.6\%.
Moreover, a line of research such as Self-Instruct [2] and Alpaca [3] demonstrates that fine-tuning models using LM-generated instructions can significantly enhance their performance. This demonstrates that under appropriate prompts, LMs are capable of generating creative content instead of replicating text encountered during training.

Nevertheless, we provide an analysis of the potential data leakage in questions generated by LM on our LMExamQA dataset. The ideal method to verify whether an LLM has encountered similar questions during training is to investigate its training data. However, the training data for most language models, often sized at several terabytes, is not openly accessible. Consequently, we investigate two primary sources of training data instead --- web data and public datasets.
Considering web data, we perform Google search on 100 randomly sampled questions from LMExamQA and retrieve the most similar web queries.
For public datasets, we look into the Flan collection [4], which contains 1,836 tasks and 15M instances, and is widely used during instruction fine-tuning for LLMs. For each question in LMExamQA, we implement a BM25 search over the inputs from the Flan collection.
Subsequently, we compute the ROUGE-L score between the question and the retrieved query, the average results of which are reported in the following table:

Web data | Public datasets
------------ | -----------------
0.293        | 0.046

The low similarity scores in both data sources imply that the majority of the questions generated by the models are not present in the two sources. We also show 3 random questions along with their corresponding queries retrieved from web data.

Question | Retrieved query
------------ | -----------------
What is the significance of tarot cards in occult practices, and how have they been adopted in modern spirituality?        | What is the significance of tarot card reading?
What are some benefits to local economies from hosting swap meets and outdoor markets? | How do farmers markets help the local economy?
How has the rise of e-commerce impacted the grocery and food retail industry? | How has e-commerce changed the retail industry?

In conclusion, this study implies that **most of the model-generated questions in LMExamQA have not been previously seen in training data**.
Besides, human-collected, fixed datasets falls short in adapting to future scenarios where more pre-training data, even including data that overlaps with existing datasets, is used.
In contrast, our paper offers not merely a single dataset but an automated pipeline for benchmark construction.
With our approach, overlap in the dataset can be seamlessly addressed by re-generating the dataset via more sophisticated prompt engineering.

We will make sure to add this study in the camera ready version.

[1]. Radford A, Wu J, Child R, et al. Language models are unsupervised multitask learners[J]. OpenAI blog, 2019, 1(8): 9.

[2]. Wang Y, Kordi Y, Mishra S, et al. Self-instruct: Aligning language model with self generated instructions[J]. arXiv preprint arXiv:2212.10560, 2022.

[3]. Taori R, Gulrajani I, Zhang T, et al. Alpaca: A strong, replicable instruction-following model[J]. Stanford Center for Research on Foundation Models. https://crfm.stanford.edu/2023/03/13/alpaca.html, 2023, 3(6): 7.

[4]. Longpre S, Hou L, Vu T, et al. The flan collection: Designing data and methods for effective instruction tuning[J]. arXiv preprint arXiv:2301.13688, 2023.

---

### Author Response · Authors · 2023-08-21
**Looking forward to your feedback**

Dear reviewers,

Thank you again for your helpful reviews. As the end of the discussion period draws near, we would like to ensure that we have adequately addressed all your concerns. If you have any further feedback, please do not hesitate to let us know.

---

### Comment · Area_Chair_S38C · 2023-08-24
**Reminder for discussion**

Dear reviewers,

I would like to remind all of you to actively participate in the author-reviewer discussion phase (which ends on Aug. 29). Active discussion will help authors better revise their work and the reviewers can better understand and evaluate the paper. Please do participate in this phase and reply to authors' rebuttal on time.

Thanks

AC

---

### Decision · Program_Chairs · 2023-09-22

**Decision:**

Accept (Poster)

**Comment:**

This paper studies the evaluation of foundation models by solving its two challenges: testing leakage and evaluation automation. The proposed framework, language-model-as-an-evaluator, enables LM as a knowledgeable examiner that allows for effortless extensibility to adopt multiple LLMs. They further designed three strategies and provided some benchmarks and analysis.

Three reviewers viewed this paper, where two of them gave acceptance ratings (scores are 7 and 8) and the other leaned towards rejection (with score 5, below the acceptance bar). The AC read the paper carefully and checked all the reviews.

Strong points for this paper are
- Using LLM as examiners is interesting and the framework is extensible since it allows multiple LLMs.
- The paper provides a dataset and benchmarking results, enhancing reproducibility and further research in the field.
- The paper also proposes a new decentralized peer examination to better understand the biases in any single examiner.

Weak points for this paper are
- The practical utility of the peer-examination system is questionable, as there may not always be multiple models available for simultaneous evaluation.
- Performance misleading: The Vicuna-13B and ChatGPT models exhibited almost perfect performance on the newly presented dataset.
- Motivation: It would be great if the paper also explained how the authors ensure the LM-generated questions are new and not influenced by testing leakage.

The AC thinks there is a lot of room for improvement in this paper. Thus, a borderline accept. But the authors strongly recommended to make changes to their camera-ready version.